# Mechanism of HMBR in Reducing Membrane Fouling under Different SRT: Effect of Sludge Load on Microbial Properties

**DOI:** 10.3390/membranes12121242

**Published:** 2022-12-08

**Authors:** Ying Yao, Yanju Wang, Qiang Liu, Ying Li, Junwei Yan

**Affiliations:** 1Foreign Environmental Cooperation Center, Ministry of Ecology and Environment of the People’s Republic of China, No. 5 Houyingfang Hutong, Xicheng District, Beijing 100035, China; 2Xuzhou Dazhong Water Operation Co., Ltd., No. 128 Heping Road, Yunlong District, Xuzhou 221000, China; 3School of Environmental Engineering, Xuzhou University of Technology, No. 2 Lishui Road, Yunlong District, Xuzhou 221111, China

**Keywords:** hybrid membrane bioreactor, membrane fouling, extracellular polymeric substances, sludge retention time, sludge loading

## Abstract

Extracellular polymeric substances (EPS) are the main causative agents of membrane fouling, and the use of a hybrid membrane bioreactor (HMBR) can mitigate this by reducing the EPS content. Four bench scale sets of HMBRs were used simultaneously to treat domestic wastewater. The effect of sludge retention times (SRT) on membrane fouling in HMBRs and the underlying mechanism were investigated by comparing and analyzing the changes in sludge load, microbial characteristics, EPS distribution characteristics, and transmembrane pressure under different SRTs. Results revealed that, among the four SRTs (10 d, 20 d, 30 d, and 60 d), the best removal rates of chemical oxygen demand and total nitrogen were observed for an SRT of 30 d, with average removal rates of 95.0% and 57.1%, respectively. The best results for ammonia nitrogen and total phosphorus removal were observed at an SRT of 20 d, with average removal rates of 84.3% and 99.5%, respectively. SRT can affect sludge load by altering the biomass, which significantly impacts the microbial communities. The highest microbial diversity was observed at an SRT of 30 d (with a BOD sludge load of 0.0310 kg/kg∙d), with Sphingobacteriales exhibiting the highest relative abundance at 19.6%. At this SRT setting, the microorganisms produced the least amount of soluble EPS and loosely bond EPS by metabolism, 3.41 mg/g and 4.52 mg/g, respectively. Owing to the reduced EPS content, membrane fouling was effectively controlled and the membrane module working cycle was effectively enhanced up to 99 d, the longest duration among the four SRTs.

## 1. Introduction

The use of membrane bioreactors (MBRs) exhibits advantages of excellent effluent quality, small footprint, separation of hydraulic retention time (HRT) and sludge retention time (SRT), and low residual sludge; however, membrane fouling has severely restricted the widespread application of this technology [1,2,3]. Generally, membrane fouling mainly refers to the adsorption and deposition of macromolecules, such as colloidal particles, sludge flocs, and organic/inorganic solutes, present in the feed solution on the membrane surface or inside its pores during the operation of MBRs, eventually leading to decreased membrane flux [4]. To restore the membrane flux, it is necessary to frequently turn the aeration on and off, backflush, and clean the membrane, which is an extremely complicated procedure with high operating costs.

Extracellular polymeric substance (EPS), recognized as a major contributing factor to membrane fouling, is generally classified into two types: soluble EPS (S-EPS), a viscous polymer that exists loosely as a liquid phase in a colloidal or dissolved state; and bound EPS (B-EPS), a vesicular polymer attached to the cell wall [5,6]. Depending on the degree of adhesion, B-EPS appears as a rheological bilayer, with a tightly bound EPS (TB-EPS) in the inner layer and loosely bound EPS (LB-EPS) on the outer layer.

B-EPS has an important influence on the physical properties of activated sludge. Forstor et al. [7] concluded that increased B-EPS content resulted in increased zeta potential of the sludge surface and deteriorated flocs, owing to increased mutual repulsion. Liu et al. [8] demonstrated that with a decrease in B-EPS content, the flocculation and settling properties of activated sludge as well as the average sludge particle size increased, and the floc structure was denser. As changes in the physical properties of activated sludge can affect membrane filtration performance, EPS is strongly correlated with membrane fouling. According to Lee et al. [9], EPS is the main contributor to membrane resistance and the decrease in membrane flux is mainly owing to the combined effect of EPS adhering to membrane pores and surfaces, and fine sludge particles. Chang et al. [10] found that a 40% reduction in EPS content was associated with a similar reduction in the resistance of the cake layer, with EPS demonstrating a significant linear relationship with membrane fouling. Notably, the EPS mentioned in the early research results mainly refers to B-EPS or total EPS. In recent years, a few scholars have started paying attention to the effects of TB-EPS and LB-EPS on membrane fouling. Jin et al. [11] suggested that the increase in LB-EPS content increased the roughness of the sludge floc interface, leading to poor sludge settling performance. Wang et al. [12] concluded that LB-EPS had a significantly greater impact on membrane fouling than TB-EPS, with a 42% reduction in its content and a corresponding 57% reduction in cake layer resistance. The effect of LB-EPS on cake layer resistance was mainly achieved by improving the physical properties associated with the activated sludge.

The significant impact of S-EPS on membrane fouling has only recently received attention; the fouling mechanism mainly involves the adsorption of small organic molecules into membrane pores, causing a reduction in their size or clogging, while macromolecules are deposited on the membrane surface, forming a gel layer. In addition, S-EPS deteriorates the filterability of the mixture, and Huang et al. [13] found that the accumulation of S-EPS in MBRs could inhibit microbial metabolic activity and reduce the membrane flux. Kimura et al. [14] concluded that the relative molecular mass distribution of S-EPS had a more significant effect on membrane fouling than its concentration or feed viscosity.

SRT is one of the important factors affecting membrane fouling. Changes in SRT affect the viscosity, zeta potential, species and distribution of microorganisms, and EPS of the mixture, ultimately affecting the membrane filtration resistance [15]. Tian et al. [16] found that a longer SRT resulted in highly stable sludge flocs, whereas the opposite led to higher EPS content. Trussell [17] found that shortening the SRT from 10 d to 2 d increased membrane fouling by 10-fold. However, Yamammoto [18] concluded that longer SRT also resulted in higher mixed liquor suspended solids (MLSS), which increased the membrane filtration resistance. The above findings are probably in conflict because of the different sludge loads, so the effect of SRT on membrane fouling should be further investigated.

In the previous research, it was found that a hybrid membrane bioreactor (HMBR), which uses both activated sludge and biofilm, could effectively control membrane fouling [19]. It looked at the mechanism of membrane fouling control, and concluded that the biofilm could improve the physical properties, such as flocculation and settleability of the activated sludge, by reducing B-EPS, especially LB-EPS, and thus improve the filtration performance of the cake layer [4,12]. EPS is derived from microbial metabolism and is greatly influenced by environmental factors and microbial species. Some scholars have studied MBR from the perspective of microorganisms, and most believed that proteobacteria was the dominant phylum of MBR [20,21]. SRT has important effects on microbial properties in MBR [22,23]. However, there were few studies about microbial properties in HMBR. We have studied the microbial community structure in HMBR and explored the principles of membrane fouling control in HMBR from the perspective of microbial properties [24]. Having compared the EPS in HMBR under different SRT conditions, we found that membrane fouling increased with prolonging SRT [25]. However, the wastewater quality used in the experiment was too volatile (with a COD of 23.9–356.2 mg/L), so the results needed to be further verified. The causes of SRT affecting membrane fouling have not yet been analyzed in depth, such as from the perspectives of sludge load.

In this experiment, a bench scale HMBR-based domestic wastewater treatment setup was conducted using four different SRTs (10 d, 20 d, 30 d, and 60 d) to clarify the EPS distribution characteristics, and microbial community structure and diversity in the HMBR under different SRT conditions. The effect of SRT on membrane fouling and its mechanism was studied from the perspective of sludge load.

## 2. Materials and Methods

### 2.1. Experimental Setup

The HMBR setup used in the study mainly comprised a feed pump, aeration tank, biological medium, membrane module, peristaltic pump, air diffuser, and an air compressor (Figure 1) with a processing capacity of 10 L/h.

The effective volume of the aeration tank was 100 L. The basin was filled with polyethylene biological media, which was a cylinder with 25 mm diameter and 12 mm height with a porous inner body, packing density of 100 kg/m^3^, specific surface area of 500 m^2^/m^3^, and a dosing volume ratio of 50%. The membrane module was a reinforced polyvinylidene fluoride (PVDF) hollow fiber microfiltration membrane with an area of 1 m^2^, pore size of 0.1 um, flux of 10 L/m^2^∙h, and HRT of 10 h. The membrane module used a peristaltic pump to pump out effluent, and the operation mode followed an “8 min on, 2 min off” sequence. Transmembrane pressure (TMP) was monitored using a vacuum gauge, and when it reached 0.1 MPa, the membrane module was taken out and was cleaned with tap water and then 5‰ NaCLO and 5% HCL solution to remove foulants on the membrane surface [26]. The air was transported into the aeration tank through the air compressor. The air diffuser was a microporous disc-shaped diaphragm with a diameter of 215 mm, the material of Ethylene-Propylene-Diene Monomer (EPDM), service area of 0.25–0.5 m^3^/unit, the bubble diameter of 0.9–1.0 mm. It served two purposes: first, to provide dissolved oxygen (DO) to the mixture to fulfil the requirements of microbial metabolism; second, to provide rising air bubbles to flush the cake layer on the surface of the membrane to reduce membrane fouling.

### 2.2. Raw Water Quality

A wastewater treatment plant in Xuzhou City, China, was used as the test site. The plant has a treatment capacity of 30,000 m^3^/d and uses domestic wastewater as the influent. The raw wastewater used in the test was obtained from the effluent of the cyclone grit chamber of the plant, and the relevant raw wastewater qualities are presented in Table 1.

### 2.3. Test Method

Four HMBR setups operated simultaneously. The sludge to be used for inoculation was taken from the mixed liquor of aeration tank of the wastewater treatment plant mentioned above, and the SRT of each setup was controlled at 10 d, 20 d, 30 d, and 60 d by discharging a certain amount of mixed liquor in the aeration tank. The DO concentration of the mixed liquor was controlled at approximately 1.0 mg/L. The effect of SRT on the removal of organic matter and nutrients was investigated by testing the water qualities at the inlet and outlet of the test setups. The effect of SRT on membrane fouling was analyzed by focusing on the changes in TMP and calculating various membrane filtration resistances. Various EPS contents were also tested to study the effects of SRT on EPS distribution characteristics, while the microbial community structures in the activated sludge were tested to analyze their diversity and the effect of EPS on their characteristics. The experiment period exceeded 6 months, from May to November 2021. Water samples were collected from the influent, mixed liquor and effluent 2 or 3 times per week. Based on the findings, the mechanism underlying the effect of SRT on membrane fouling in HMBR was elucidated.

### 2.4. Analytical Method

Samples were collected from the influent, mixed liquor and effluent 2 or 3 times per week. MLSS and mixed liquor volatile suspended solids (MLVSS) were analyzed according to gravimetric method. COD was analyzed according to potassium dichromate method. BOD_5_ was analyzed according to dilution inoculation method. NH_4_^+^-N was analyzed according to Nessler’s reagent spectrophotometry. TN was analyzed according to potassium persulfate oxidation—ultraviolet spectrophotometry. TP was analyzed according to Mo-Sb anti spectrophotometry [27]. To measure the biofilm concentration, 25 carriers were carefully collected from the aeration tank and dried for 2 h at 105 °C. The biofilm weight was obtained by subtracting the original weight of the clean carriers from the dried ones. The biofilm concentration was finally expressed as mg/L by considering the total number of carriers and the total volume of the aeration tank [19]. EPS were extracted according to Wang et al. [4] S-EPS, LB-EPS and TB-EPS were determined as COD per gram MLVSS. Total membrane resistance (*R_t_*), membrane intrinsic resistance (*R_m_*), pore blocking resistance (*R_p_*), and cake layer resistance (*R_c_*) were determined following Darcy’s law [28], which can be expressed as:(1)$$J=\frac{\Delta P}{\mu R_{t}}=\frac{\Delta P}{\mu \left(R_{m}+R_{p}+R_{c}\right)}$$
where *J* = membrane flux; ∆*P* = TMP; *μ* = absolute viscosity.

The microbial community was detected using high-throughput sequencing (HTS) technology [24]. The specific method of HTS was introduced as follows:(1)DNA extraction. The DNA was extracted from the samples using the MoBio PowerSoil DNA extraction kit (MO BIO Laboratories, Carlsbad, CA, USA) following the manufacturer’s instructions.(2)Polymerase chain reaction (PCR) amplification. PCR amplification of 16S rRNA genes was performed using general bacterial primers 515F (5′-GTGCCAGCMGCCGCGGTAA-3′) and 926R (5′-CCGTCAATTCMTTTGAGTTT-3′). The primers also contained the Illumina 5’overhang adapter sequences for two-step amplicon library building.(3)Miseq HTS. The barcoded PCR products were purified using a DNA gel extraction kit (Axygen, Shanghai, China) and quantified using the FTC-3000 TM real-time PCR. The libraries were sequenced by 2 × 300 bp paired-end sequencing on the MiSeq platform using MiSeq v3 Reagent Kit (Illumina) at TinyGene Bio-Tech (Shanghai) Co., Ltd., China.(4)Bioinformatic analysis. The raw fastq files were demultiplexed based on the barcode. Paired-end (PE) reads for all samples were run through Trimmomatic (version 0.35) to remove low-quality base pairs. Trimmed reads were then merged using FLASH program (version 1.2.11). The low quality contigs were removed based on screen.seqs command in mothur (version 1.33.3). The cleaned reads were clustered at 97% sequence identity into operational taxonomic units (OTUs) using the UPARSE pipeline (usearch version v8.1.1756). The OTU representative sequences were assigned for taxonomy against Silva 128 database by the classify.seqs command in mothur. Taxonomies (from phylum to species) of the OTUs were determined according to the National Center for Biotechnology Information. Based on the taxonomy, the statistical analysis of community structure was carried out at the level of phylum, class, order, family, genus and species.

## 3. Results and Discussions

### 3.1. Effect of SRT on HMBR Treatment Efficacy

During the experiment, SRT had no significant effect on biofilm concentration, which was essentially maintained at approximately 1208 mg/L. Changes in the activated sludge concentration under different SRT conditions are presented in Table 2. Although both MLSS and MLVSS increased with increasing SRT, the increase in MLSS was obviously higher than that in MLVSS. When SRT exceeded 30 d, the ratio of MLVSS/MLSS decreased significantly, indicating a significant reduction in the sludge activity. When SRT was too long, MLVSS increased, sludge load was greatly reduced, and microbial metabolism was inhibited, which must be the main reason for the decrease in microbial activity.

The effectiveness of HMBR in removing the COD, NH_4_^+^-N, TN, and TP under four different SRT conditions is presented in Table 3.

Effective removal of organic matter is related to the biomass [29]. In this experiment, HMBR effectively removed the organic matter under different SRT conditions, with the average COD removal rate exceeding 90%. Extending the SRT from 10 to 20 d resulted in an increase in MLVSS and COD removal rate by 159 mg/L and 2.9%, respectively. A further increase in SRT from 20 to 30 d caused the MLVSS and COD removal rate to increase by 1095 mg/L and 1.2%, respectively. When the SRT was extended to the maximum duration (60 d), although MLVSS increased by 221 mg/L, the COD removal rate increased only by 0.1%, which may be related to the substantial decrease in sludge activity.

The removal of NH_4_^+^-N relies mainly on nitrifying bacteria, which have a long generation cycle and slow reproduction rate [30]. Therefore, when the SRT was extended from 10 to 20 d, the NH_4_^+^-N removal rate increased from 97.6 to 99.5%. The NH_4_^+^-N removal rate remained stable in further SRT extensions, indicating that 20 d SRT can satisfactorily fulfil the requirements of nitrifying bacteria for their generation cycle.

Biological removal of nitrogen requires two processes: nitrification under aerobic conditions and denitrification under anoxic conditions. Similarly, biological phosphorus removal requires two processes: phosphorus release reaction under anaerobic conditions, and phosphorus absorption reaction under aerobic conditions. The traditional theory of biological removal of nitrogen and phosphorus owes to different reaction conditions, and different reactions must be performed in different structures. However, some scholars [8,31] argued that at low DO concentrations, anoxic zones or even anaerobic zones may be formed inside activated sludge of a certain particle size or inside biofilms of a certain thickness owing to oxygen mass transfer resistance, causing simultaneous nitrification and denitrification. In addition, owing to the effect of DO on mass transfer resistance, aerobic, anoxic, and anaerobic zones may be formed in different areas in the same aeration tank, where nitrification, denitrification, phosphorus release, and phosphorus absorption reactions occur simultaneously. In this experiment, the DO concentration was controlled at a low level of approximately 1.0 mg/L and the reactor exhibited a certain nitrogen and phosphorus removal function. The highest TN removal rate was observed at an SRT of 30 d, with a mean value of 57.1%, while the highest TP removal rate was observed for an SRT of 20 d, with a mean value of 84.3%. TN and TP removal rates at the SRT of 60 d both decreased significantly, which may be due to the lack of organic carbon sources as a result of the low sludge load. Due to of insufficient organic carbon sources, the denitrification process was greatly affected, and the metabolism of polyphosphorus bacteria was inhibited.

### 3.2. Effect of SRT on EPS Distribution Characteristics

As EPS is a metabolic product of microorganisms, microbial community structure and diversity have a very important impact on EPS content.

In this experiment, the total EPS first exhibited a decreasing then increasing trend with increasing SRT. When the SRT was extended from 10 to 20 d, the total EPS (mean value) decreased from 25.71 mg/g to 24.08 mg/g, by 6.34%. At an SRT of 30 d, the total EPS decreased again by 0.02 mg/g, and the mean concentration was 24.06 mg/g. At the highest SRT of 60 d, the total EPS increased rapidly, and the mean concentration reached 26.96 mg/g. Among the three EPS types, S-EPS concentration exhibited a different trend from that of total EPS. An increase in SRT from 10 to 20 d caused a decrease in S-EPS by 6.81%, from 3.67 mg/g to 3.42 mg/g. When SRT was extended to 30 d, it decreased again by 0.01 mg/g, with an average concentration of 3.41 mg/g. However, no significant change in S-EPS was observed when the SRT was extended up to 60 d. Although both LB-EPS and TB-EPS concentrations first exhibited a decreasing then an increasing trend, the minimum value corresponded to different SRTs, with the least amounts of LB-EPS and TB-EPS observed at SRTs of 30 d and 20 d, respectively (Figure 2).

### 3.3. Effect of SRT on Membrane Fouling

The changes in TMP were significantly different under the four different SRT conditions. For SRT ≤ 30 d, the increase in TMP gradually levelled off with increasing SRT, with a corresponding increase in the working cycle of the membrane module. When the SRT reached 60 d, TMP increased rapidly, with a significant reduction in the working cycle of the membrane module. This must be caused by both S-EPS and LB-EPS. S-EPS can affect membrane pore resistance, and LB-EPS has a significant relationship with cake layer resistance [4]. Therefore, reducing both S-EPS and LB-EPS can improve the membrane filtration performance. Otherwise, it may be related with the distribution of EPS, especially the ratio of S-EPS to LB-EPS. The ratio had a maximum value of 0.7544 at SRT of 30 d, and a minimum value of 0.5007 at SRT of 60 d. Therefore, it can be inferred that a lower ratio of S-EPS to LB-EPS can lead to more serious membrane fouling. Furthermore, a more detailed analysis showed that with the rapid increase in TMP at SRT of 60 d due to the increase in LB-EPS, the effect of LB-EPS on membrane fouling is obviously greater than that of S-EPS. For the four SRT periods, the working cycles of the membrane module were 57 d, 74 d, 99 d, and 43 d, respectively (Figure 3). Using the working cycle of the membrane module for an SRT of 60 d as the base, the TMP at the same SRT was calculated as 0.1 MPa (point A) when operating up to the 43rd day, while those for SRTs of 10 d, 20 d, and 30 d were 0.08 MPa (point B), 0.06 MPa (point C), and 0.053 MPa (point D), respectively. The *R_t_* at an SRT of 10 d, 20 d, and 30 d were 20%, 40%, and 47% lower, respectively, than that at an SRT of 60 d. According to the results, the least membrane fouling was observed at an SRT of 30 d.

The calculated BOD sludge loads were 0.0475 kg/kg∙d, 0.0428 kg/kg∙d, 0.0310 kg/kg∙d, and 0.0259 kg/kg∙d for SRTs of 10 d, 20 d, 30 d, and 60 d, respectively. Therefore, it can be concluded that 0.0310 kg/kg∙d was the optimum sludge load for controlling membrane fouling.

### 3.4. Effect of SRT on Microbial Community Structure

Variations in SRT can change the sludge load, which has an important influence on the growth and reproduction of microorganisms and their composition. The microbial community structure at class level in activated sludge under different SRT conditions is illustrated in Figure 4. Analysis revealed that Sphingobacteriia were the most abundant microorganisms at an SRT of 10 d, with a relative abundance of 39.0%. The most abundant microorganisms observed at SRTs of 20 d, 30 d, and 60 d were all Betaproteobacteria with relative abundances of 12.1%, 24.3%, and 20.9%, respectively.

Proteobacteria and Bacteroidetes are the main heterotrophic bacteria in activated sludge. They can take organic matter as the carbon source and energy, decompose organic pollutants and synthesize their own cell substances, and play a major role in the removal of COD [21,32]. Sphingobacteriia belongs to Bacteroidete, and is more suitable for survival in an anaerobic microenvironment [33]. A higher abundance of Sphingobacteriia was detected under all four SRT conditions, with the first dominant bacteria under SRT of 10 d, and the second dominant bacteria under SRT of others, which confirmed that there must be an anaerobic microenvironment in the aeration tank from the side. When the SRT was extended over 20 d, the reason why Betaproteobacteria became the first dominant bacteria should be related to the reduction in sludge load. At lower sludge load, Betaproteobacteria had better competitiveness for organic substances than Sphingobacteriia.

Some bacteria are able to secrete more EPS in their metabolism than others, e.g., Betaproteobacteria [34,35]. At an SRT of 30 d, the higher the abundance of Betaproteobacteria, the more abundant filamentous bacteria were as well, as EPS contributed to the biofilm generation of the medium and the membrane surface. The participation of biofilm in biodegradation can effectively reduce LB-EPS [4], and membrane fouling was controlled effectively.

### 3.5. Effect of SRT on Microbial Alpha Diversity

A rarefaction curve is used to extract a random number of individuals from a sample, count the number of species represented by these individuals, and construct curves with the number of individuals and the number of species. It can be used to compare the richness of species in samples with different amounts of sequencing data, and also to show whether the amount of sample sequencing data is reasonable. Random sampling of sequences is used to construct a rarefaction curve, with the number of sequences extracted and the number of Operational Taxonomic Units (out) they can represent. When the curve becomes flat, it indicates that the sequencing data quantity is reasonable [36], and that more data only produce a small amount of noutOTU. Therefore, the sequencing depth of the sample can be obtained by making a rarefaction curve.

Alpha diversity can be used to reflect the abundance and diversity of microbial communities by analyzing the Chao, Ace, Shannon, and Simpson indices. In this experiment, the bacterial gene library coverage of sludge samples under every SRT all exceeded 99.5%, indicating that most of the microorganisms were detected and the sequencing results were highly reliable. When the SRT was extended from 10 d to 30 d, the Chao, Ace, and Shannon indexes all increased substantially to 1078.4, 1070.8, and 5.43, respectively, but the Simpson index had a significant drop to 0.0126. Extending the SRT up to 60 d, the Chao and Ace indexes grew slightly by 0.12% and 1.20%, respectively, and the Shannon index had a smaller decline of 1.47%. However, the Simpson index showed a significant increase, by 21.43% (Figure 5). The higher the Chao, Ace, and Shannon indices and the smaller the Simpson index are, the higher the abundance of the species in the sample is [24]. The analytical results clearly indicated the highest microbial diversity at an SRT of 30 d, followed by that at 60 d, then 20 d, and the worst diversity at an SRT of 10 d. In the previous study, we concluded that the best Alpha diversity for the activated sludge at the SRT was 10 d, followed by 20 d and 30 d. This conclusion is quite opposite to the results of this experiment, and the reason should be related to the low organic concentration of the raw wastewater (COD < 20 mg/L) in the previous experiment. The organic concentration of raw wastewater was too low, resulting in a sludge load too small. After calculations, the COD sludge load with an SRT of 10 d in the previous experiment was only 0.0897 kg/kg·d, which was likely to be roughly equal to the BOD sludge load when the SRT was 30 d in this experiment.

### 3.6. Mechanism of SRT Action on Membrane Fouling in HMBR

SRT can change the sludge load by affecting the biomass. With gradually increasing SRT, the biomass gradually increased and the sludge load decreased accordingly. The total biomass in the reactor was 5203 mg/L, 5781 mg/L, 7973 mg/L, and 9536 mg/L at SRTs of 10 d, 20 d, 30 d, and 60 d, respectively, corresponding to BOD sludge loads of 0.0475 kg/kg∙d, 0.0428 kg/kg∙d, 0.0310 kg/kg∙d, and 0.0259 kg/kg∙d, respectively. As the sludge load is one of the important factors for microbial growth and reproduction, there must be an optimum sludge load with a particular microbial community composition which results in the least amount of EPS produced by microbial metabolism. According to the results, the microbial community exhibited the greatest abundance at a BOD sludge load of 0.0310 kg/kg∙d (corresponding to 30 d SRT), with Betaproteobacteria being the largest number of microbial species at the class level with a relative abundance of 24.3%. At this point, the levels of S-EPS and LB-EPS, which are reportedly the major contributors to membrane fouling, were the lowest at 3.41 mg/g and 4.52 mg/g, respectively. TMP growth slowed down significantly owing to the decrease in EPS content. The membrane module working cycle lasted 99 d for an SRT of 30 d, the longest duration among the four SRTs, indicating the least membrane fouling under this condition.

## 4. Conclusions

SRT had an important effect on the removal of organic substances and nutrients, and the membrane filtration performance of the HMBR. The removal effect of COD and NH_4_^+^-N increased with the increase in SRT, and the average removal rates were 95.1% and 99.8% for an SRT of 60 d. The best TN removal effect was achieved at SRT of 30 d, and the best TP removal effect was achieved at an SRT of 20 d, with average removal rates of 57.1% and 84.3%, respectively. The membrane was least contaminated at an SRT of 30 d, with a module filtration period of 99 d. The membrane was most contaminated at an SRT of 60 d, with a module filtration period of 43 d. The microbial diversity was the highest at an optimal sludge load value (0.0310 kg/kg∙d), with Betaproteobacteria as the first dominant bacteria (24.3% abundance), and Sphingobacteriia the second dominant (19.6% abundance). Both bacteria were able to decompose organic substances in water and synthesize their own cellular material. Among them, Betaproteobacterium was able to secrete more EPS than the other bacteria, and could promote the formation of biofilm when it was higher in abundance. Participation of biofilm in biochemical reactions can effectively reduce the content of EPS, especially S-EPS and LB-EPS. Reduction in LB-EPS content will improve the flocculation and sedimentation of activated sludge, and thus improve the filtration performance of the filter cake layer. S-EPS may be related to the filtration resistance of membrane pores. In addition, EPS distribution may have a greater impact on membrane contamination, and a lower S-EPS/LB-EPS ratio can trigger more serious membrane contamination, as LB-EPS has a significantly greater impact on membrane contamination than S-EPS.

## Figures and Tables

**Figure 1 membranes-12-01242-f001:**
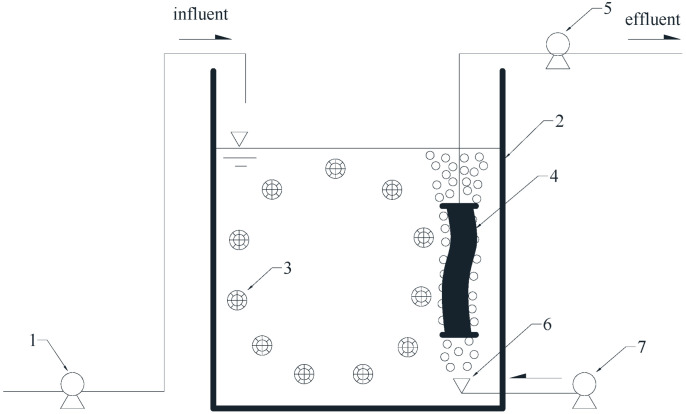
Schematic diagram of the HMBR setup. 1 feed pump; 2 aeration tank; 3 biological medium; 4 membrane module; 5 peristaltic pump; 6 air diffuser; 7 air compressor.

**Figure 2 membranes-12-01242-f002:**
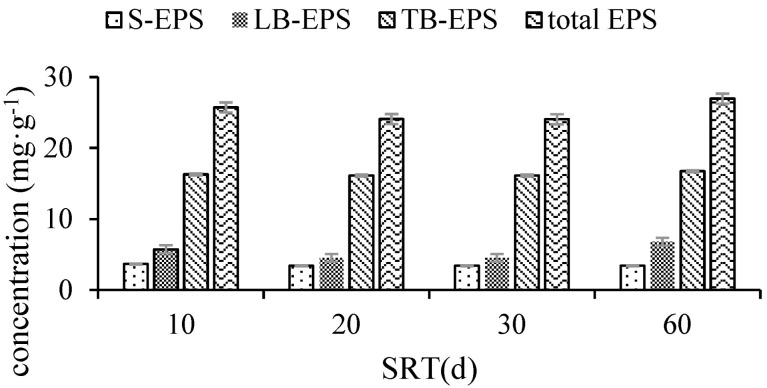
EPS distributions under the four different SRT conditions.

**Figure 3 membranes-12-01242-f003:**
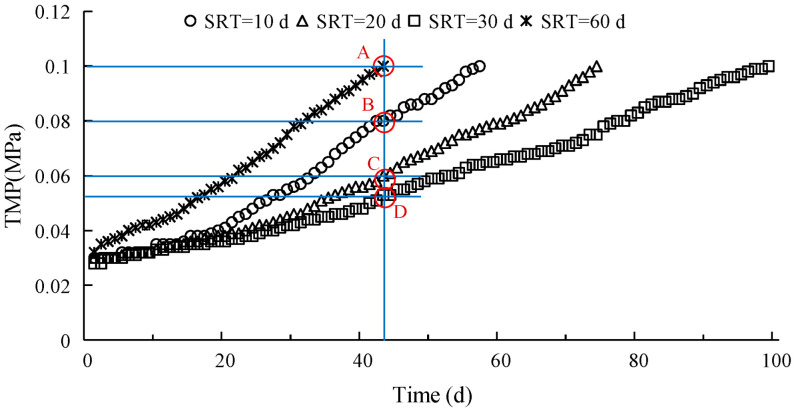
TMP variations under the four different SRT conditions.

**Figure 4 membranes-12-01242-f004:**
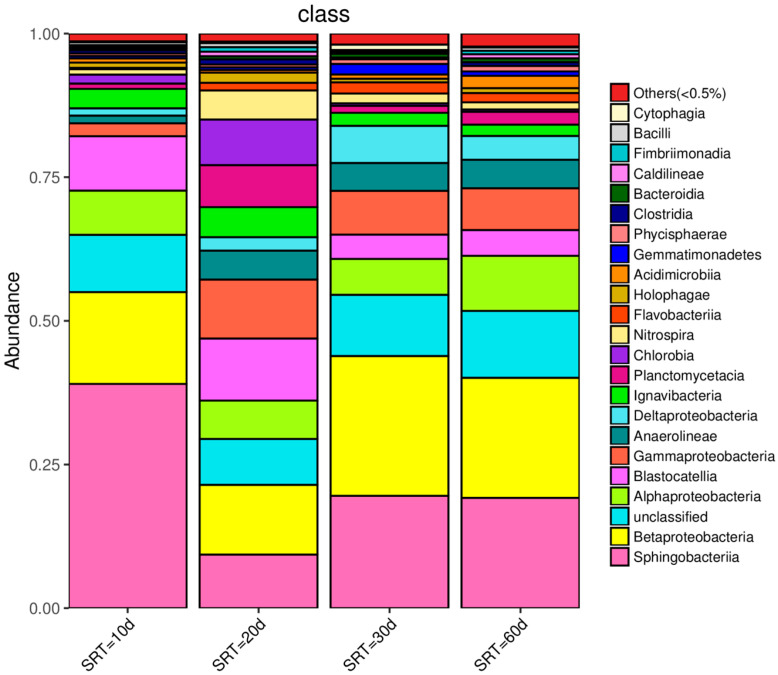
Microbial community structure in activated sludge under different SRT conditions.

**Figure 5 membranes-12-01242-f005:**
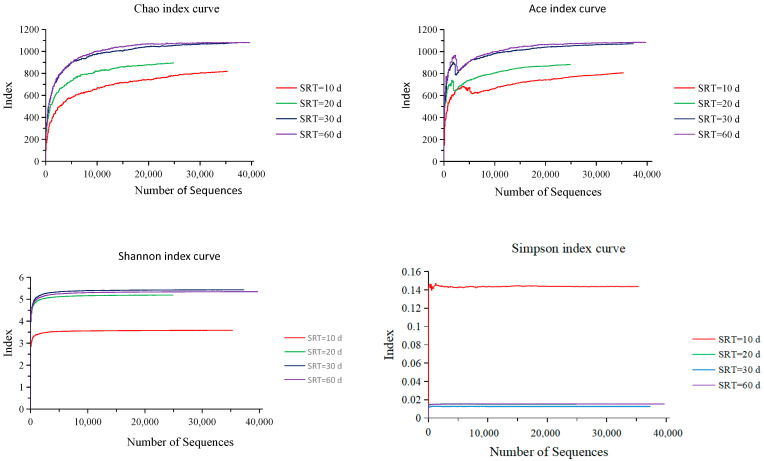
Microbial alpha diversity in activated sludge under the four different SRT conditions.

**Table 1 membranes-12-01242-t001:** Characteristics of the raw wastewater.

Parameter	Description	Average
Chemical oxygen demand (COD) (mg/L)	257–386	286
Biochemical oxygen demand 5-day test (BOD_5_) (mg/L)	95–145	103
Ammonia nitrogen (NH_4_^+^-N) (mg/L)	36.5–47.9	41.3
Total nitrogen (TN) (mg/L)	48.3–53.3	50.8
Total phosphorus (TP) (mg/L)	4.5–5.7	5.1
Temperature (°C)	17.8–22.6	21.3
pH	7.53–7.69	7.61

**Table 2 membranes-12-01242-t002:** Activated sludge concentration under different SRT conditions.

SRT (d)	MLSS (mg/L)	MLVSS (mg/L)	MLVSS/MLSS
10	3871–4053 (3995)	2201–2533 (2328)	0.56–0.61 (0.58)
20	4412–4693 (4573)	2359–2690 (2487)	0.52–0.56 (0.54)
30	6690–6813 (6765)	3423–3769 (3582)	0.51–0.55 (0.53)
60	8250–8468 (8328)	3675–3953 (3803)	0.44–0.47 (0.46)

Note: Values in brackets are the average, n > 30.

**Table 3 membranes-12-01242-t003:** Organic and nutrients removals by HMBR under different SRT conditions.

SRT (d)	Sample	COD (mg/L)	NH_4_^+^-N (mg/L)	TN (mg/L)	TP (mg/L)
	Influent	257–386 (286)	36.5–47.9 (41.3)	48.3–53.3 (50.8)	4.5–5.7 (5.1)
10	Effluent	19.1–28.8 (26.0)	0.8–1.5 (1.0)	20.3–28.6 (25.7)	0.8–1.3 (1.2)
	Removal (%)	90.9 ± 2.1	97.6 ± 0.8	49.4 ± 2.3	76.5 ± 2.1
20	Effluent	16.1–27.8 (17.7)	0.1–0.5 (0.2)	18.3–25.1 (22.8)	0.5–1.1 (0.8)
	Removal (%)	93.8 ± 2.2	99.5 ± 0.5	55.1 ± 1.9	84.3 ± 2.0
30	Effluent	13.8–15.1 (14.4)	0.1–0.4 (0.2)	17.3–25.6 (21.8)	0.7–1.1 (1.0)
	Removal (%)	95.0 ± 1.8	99.5 ± 0.5	57.1 ± 1.8	80.4 ± 1.9
60	Effluent	13.6–15.3 (14.1)	0.1–0.5 (0.1)	23.8–31.4 (29.5)	1.2–1.9 (1.7)
	Removal (%)	95.1 ± 1.8	99.8 ± 0.2	41.9 ± 1.6	66.7 ± 2.2

Note: Values in brackets are the average, n > 30.

## Data Availability

Data is contained within this article.

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
