# Peer review of "Mechanism of HMBR in Reducing Membrane Fouling under Different SRT: Effect of Sludge Load on Microbial Properties"

_membranes, 2022, doi:10.3390/membranes12121242_

Round 1

Reviewer 1 Report

The manuscript “Impact and Mechanism of SRT on membrane fouling in HMBR based on EPS characterization” (Manuscript ID: membranes-2036922) by Yao et al. investigated the effect of difference in sludge retention time (SRT) on the performance of hybrid membrane bioreactor (HMBR), especially on the membrane fouling. The study consists of a sufficient amount of physicochemical (and microbial) data, and the main conclusion that operation with the HRT of 30 d showed the most preferable performance (the levels of S-EPS and LB-EPS were the lowest at 3.41 mg/g and 4.52 mg/g, respectively; increase in TMP slowed down significantly owing to the decrease in EPS content; the membrane module working cycle lasted 99 d, the least membrane fouling under this condition) was supported by the data. Yet, I would suggest the authors to revise the manuscript in several aspects.

Major comments

1. The authors may need to emphasize the significance or novelty of your research. Whereas I understood that the authors focused on the HRT, many other researchers had also focused on that point. What are the differences from those previous study? (Maybe, the use of HMBR is one of the points?) The results that HRT of 30 d is the best was as predicted or unpredicted? To describe the authors’ prediction before doing experiments and explain how the authors interpreted the results may help the emphasis of the significance. The authors may also need to compare the present research with other MBR papers focusing on HRT. It would help to emphasize the significance of the research.

2. Figure 4 shows the microbial community structures at phylum, class, order, family, genus and species levels. However, all the six panels seem not be presented in the main text. I don’t think all the panels are necessary in Fig. 4 and I think the authors should select only the panels they need in the main text. Also, the quality of Fig. 4 may be poor. The texts in the annotations for the taxon are unclear. The panel for species-level microbial community structure contains many “uncultured” ones, which are in most cases not annotated as species-level. In addition, especially for microbiologists, the results that the abundance of the “unclassified” one is lower in species-level figure than in genus-level figure could be weird. And, again, the microbial community data are not well discussed in the main text. Please reconsider Fig. 4.

3. Overall, I think the discussion is lacking.

Minor comments

1. In the main text, the word "unclassified" is used as if it were a taxon name. [e.g., Line 229]

2. [Line 285] EPS, which are reportedly the major contributors

--> EPS, which are reported as the major contributors

(Is this okay?)

3. It may be weird that "mechanism" is included in the title. It could be read as "mechanism of SRT".

Author Response

We have revised our manuscript taking in account the comments.

Reviewer 2 Report

Very good work with valuable results. Comments are following:

1.       Introduction.

The scientific background and the beneficial role of HMBR might be better justified in introduction, to underline the mechanisms of fouling mitigation providing the link to the targets of the current work.

2.       Materials and Methods

More details could be provided in part 2.1 Experimental setup about the experimental facility such as the polyethylene biological media, the type of the air diffuser and the size of air bubbles or the chemical cleaning process. In addition, information should be given about the sludge inoculum used during operation, as well as the microbial community identification by HTS in part 2.4 Analytical method.

3.       Results and Discussion

Line 154: How was biofilm concentration measured?

Lines 158-159 and Table 2: Why MLVSS/MLSS ratio reduced at the highest SRT? Which is the definition of ‘sludge activity’?

Table 3: Statistical analysis should be conducted including standard deviation in order to have a more clear identification of the experimental results trends and the corresponding patterns. The authors should justify the reduction in TN and TP removal rates observed at longer SRTs.

Part 3.2. Effect of SRT on membrane fouling and Figure 2: An explanation about the highest reduction rate of TMP at the longest SRT is missing.

Part 3.3. Effect of SRT on microbial community structure: More details should be given about the presence of certain microorganisms linked to the corresponding operation conditions.

Part 3.4: Effect of SRT on microbial alpha diversity: The scientific background behind alpha diversity and the corresponding indices should be provided in relation to TMP change and membrane fouling mechanisms.

3.6. Mechanism of SRT action on membrane fouling in HMBR: Discussion of the experimental results and the corresponding findings of SRT effect on membrane fouling is rather short; a more detailed analysis should be made relating SRTs to sludge loading, EPS content and bio community composition especially at longest SRT where TMP change increased to the highest rate.

Overall comment: work is mainly focused on the presentation of the experimental values, while appropriate justification of the mechanisms of membrane fouling related to the corresponding observations is missing.

Author Response

Please see the attchment.

Reviewer 3 Report

This is a well written manuscript. Some small comments:

1. Abstract section: use the whole term for soluble EPS and loosely bound EPs, thus avoiding adding a new definition to the abstract section. These terms and defined in the introduction, but they need a specification in the abstract, so the text can be understood as a stand-alone document.

2. L 90-94: Please rephrase so the aim is clearly established here. Avoid the use of the plural first person. As written, there is a mix between the aim and conclusion.

3. L118-121: Please indicate the period of sampling, and number of samples to get the range and average reported. Period 6 months in a year? what year?? what months??

4. L125-135: SRT refers to microorganisms residence time, but I can not find in the methodology the HRT for the four tests. And the testing period of each test. It seems from the description that microorganisms are immobilized on carriers. Please indicate the way authors make sure SRT is kept at the desired time.

5. L157-159: Please rephrase, the term significant is repeated 3 times in these two lines

6. Table 3: Please create a heading with "SRT" for the table. Transpose the tables so the headings can be simplified, no need to repeat the content "effluent" and "average removal".

7. Description of microbial analysis is needed in M&M section. It really amazes me how authors could miss adding this methodology to the manuscript when they repeatedly indicated so in the aim and abstract

8. L256-270: The differences are too small to seem significant. Please add relative distribution to the comments, seems differences observed may keep a closer relation to the S/LB EPS relative distribution.

9. Please unify conclusions in a single paragraph. Indicate clearly the conclusion of the study, no a resume of all different values reported for the different cases evaluated. 

10. After reading the whole manuscript L94-95 are not correct. Please rephrase, the mechanism was not elucidated, just an explanation is given, which in this specific case indicates a higher abundance of some organisms, but there are other several being unclassified.

Round 2

Reviewer 2 Report

Comments have been addressed and paper can be published as it is.

Reviewer 3 Report

Authors provided suitable corrections to the document.